# PIT-QMM: A Large Multimodal Model for No-Reference Point Cloud Quality Assessment

## Abstract

Large Multimodal Models (LMMs) have recently enabled considerable advances in the realm of image and video quality assessment, but this progress has yet to be translated to the domain of 3D assets. We are interested in using these models to conduct No-Reference Point Cloud Quality Assessment (NR-PCQA), where the aim is to automatically evaluate the perceptual quality of a point cloud in absence of a reference. We begin with the observation that different modalities of data – text descriptions, 2D projections, and 3D point cloud views – provide uniquely useful insights into point cloud quality. We leverage this to devise a multimodal dataset construction strategy providing a holistic combination of multiple types and levels of information. We then construct PIT-QMM, a novel LMM for NR-PCQA that is capable of consuming text, images and point clouds to predict quality scores. Extensive experimentation shows that our proposed method outperforms the state-of-the-art by significant margins on popular benchmarks with fewer training iterations, and thorough ablations validate our dataset construction strategy. Code and datasets are available at https://anonymous.4open.science/r/pit-qmm-BD1F/.

## 1 Introduction

Point clouds are collections of 3D points with color, opacity, and other features representing objects or environments. Recent advances have cemented point clouds as pivotal data structures for 3D representation, particularly in applications such as autonomous driving, immersive gaming and digital twin systems (Qi et al., 2017; Afham et al., 2022; Zhang et al., 2022a). These simple yet information-rich representations allow detailed spatial analysis with minimal assumptions about underlying geometry. However, this also means that they are highly susceptible to corruption at all stages of their operation cycle, from acquisition due to sensor inaccuracies, to compression losses and transmission errors. Not only do these impairments affect perceptual quality for humans, but they also directly impact downstream applications.

As a result of this corruption, assessing the quality of point clouds automatically and at scale has emerged has a key research question. Traditional metrics like PSNR or SSIM (Wang, 2004), widely used for image and video quality assessment, have been adapted for point clouds, but these often fail to capture the intricacies of 3D data. Learning-based methods have improved upon these metrics, but suffer from a lack of generalizability due to a scarcity of labeled training data. Most PCQA (point cloud quality assessment) datasets are only of the order of hundreds of distorted samples with labels (i.e. mean opinion scores, or MOS) due to the highly laborious data collection process. This is particularly disadvantageous in the No-Reference (NR) setup, where there are no pristine point clouds available to serve as a reference for quality assessment, leading to further scarcity of information.

The foremost solution that has emerged to attain generalizability in the face of label scarcity has been unsupervised pretraining on large datasets. In particular, approaches based on contrastive learning have risen to the top of many benchmarks, such as CONTRIQUE (Madhusudana et al., 2022), ReIQA (Saha et al., 2023) and recently in the point cloud domain, CoPA (Shan et al., 2024). However, these require large numbers of positive and negative paired samples and extensive augmentation schemes, which make them computationally intense to train even though the base datasets may be relatively small. Moreover, the design of such samples requires great thought and care, es-

pecially in the quality domain, where the fine visual low-level details of media are paramount. This leads to it being relatively difficult to apply the benefits of scaling, as observed in CONTRIQUE and ReIQA, which are trained on million-scale base datasets.

Elsewhere, foundation models trained on billion-size datasets with billions of parameters have delivered impressive performance on several benchmarks. In particular, large image-text multimodal models (LMMs) have recently been applied to image and video quality assessment with great success, as in Q-Align (Wu et al., 2023b). While these models have delivered state-of-the-art performance and generalizability in the 2D space, they do not straightforwardly extend to the 3D space. One approach is to simply take 2D projections of the 3D content and apply the model, but as we demonstrate in Section 4.5, this tends to yield sub-optimal results. This is likely because of the loss of information in the projection process due to factors such as occlusion, depth ambiguity and viewpoint dependency. There have been efforts to recover the lost information by supplementing predictions from 2D quality foundation models with traditional point cloud quality statistics, as in LMM-PCQA (Zhang et al., 2024). However, since they rely on handcrafted features to extract information from the point cloud, they cannot leverage the benefits of training to learn more sophisticated quality features, and crucially, cannot learn the interactions between different modalities of data.

Efforts have been made to develop point-text LMMs, particularly for semantic tasks such as object classification and question-answering. However, the fixed context length and quadratically-scaling computational costs of transformers have limited their application to point clouds of relatively small size and complexity i.e. on the order of tens of thousands of points. PCQA datasets usually contain point clouds with hundreds of thousands to millions of points, as fine-grained details about quality are often not perceptible at lower resolutions. As a result, when these models are fine-tuned on PCQA datasets, there is either a tremendous loss of information due to downsampling, or a large domain shift in terms of the nature of the point clouds they were pretrained to handle. This leads to sub-optimal performance on PCQA, as we demonstrate in Section 4.5.

In essence, image-text quality foundation models are able to understand quality, but not 3D content, whereas point-text multimodal models are able to understand 3D content but not at the granularity required for quality. This paper seeks to bridge this gap. To do so, we propose the Point-Image-Text Quality Multimodal Model, or PIT-QMM, which is a first-of-its-kind point-image-text LMM for PCQA. The key insight behind the construction of the model is that different modalities can provide complementary information about quality. Point cloud patches can provide information about local variations which are typically lost in the projection process. Image projections can supply a global picture about the point cloud which cannot currently be delivered by off-the-shelf point cloud encoders for large clouds. Finally, text inputs can used to describe the psychometric setup for rating collection, the details of which often change how the point cloud is perceived, and thus can affect the final quality scores. PIT-QMM is trained end-to-end and is fully differentiable, which means it may also be used as a training loss in applications where quality is a key factor for performance, such as content generation, editing or enhancement.

Our main contributions may be summarized as follows:

- We bridge the algorithmic gap between 2D and 3D quality assessment with our proposed PIT-QMM, which is the first end-to-end point-image-text multimodal model optimized for PCQA.
- We propose a carefully-designed dataset construction recipe for NR-PCQA, where each piece aims to leverage a complementary source of information relevant to PCQA.
- We perform thorough performance benchmarking, and show that our model beats state-of-the-art methods by a large margin despite requiring fewer training iterations. We also perform a thorough ablation study to validate the importance of each step in our dataset construction recipe.

## 2 RELATED WORK

Quality assessment is a key component of media processing pipelines, from images to 3D content. Traditionally, point cloud quality metrics have evaluated the distortion based on geometric discrepancies compared to a pristine reference. For example, p2point (Mekuria et al., 2016) measures the

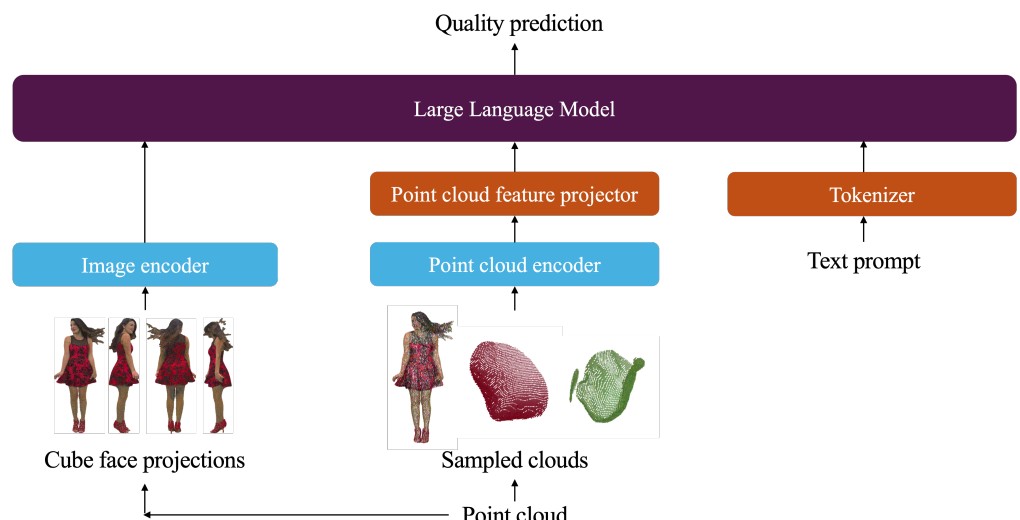

Figure 1: **An overview of the proposed Point-Image-Text Quality Multimodal Model (PIT-QMM).** PIT-QMM takes a raw point cloud and extracts both 2D and 3D views that provide different kinds of information for quality assessment. Rich feature representations of these views are encoded by pretrained foundation models. These representations are then passed into a large multimodal model along with a textual description of the task and experimental setup, which is trained to predict quality scores.

distance between corresponding points to quantify distortions. P2plane (Tian et al., 2017) improves over p2point by projecting p2point distances along the surface normal, thus taking the distance of each point from the nearest point on the cloud. Both p2point and p2plane have been included in the MPEG PCC standard (Chen et al., 2023).

However, these metrics do not adequately account for the fact that the human visual system (HVS) is highly sensitive to local structural distortions, as measured by SSIM, and is also affected by color discrepancies. Thus, several metrics that consider both geometric structure and color have been considered, such as PCQM (Meynet et al., 2020), GraphSIM (Zhang et al., 2021) and MPED (Yang et al., 2022b). For example, PCQM computes local curvature discrepancies and color inconsistencies and pools them with an optimally-weighted linear combination. On the other hand, GraphSIM also considers projections, where it renders the point cloud onto six perpendicular image planes of a cube and computes image-based quality features along with point cloud-based features. Finally, these features are aggregated via linear combinations to produce a final score.

The preceding metrics are all full-reference (FR), which means they assume that a pristine source point cloud is available. However, this may not be the case in many practical scenarios. For the no-reference (NR) case, traditional methods typically compute various handcrafted features and then train a regressor to obtain a predicted MOS score. For example, Zhang et al. (2022b) propose an NR-PCQA metric that projects point clouds into feature domains based on geometry and color and regresses the predicted MOS using a support vector machine (SVM). While hand-crafted features have the advantage of explicit semantic meaning, our understanding of point cloud distortions itself is still quite limited, which handicaps the development of features that can handle more complex distortion settings.

With the advent of deep learning, learning-based methods have instead become the leaders in NR-PCQA. PQA-Net (Liu et al., 2021) takes cubic projections and extracts multi-view features, which it then uses to jointly predict quality and distortion type. ResSCNN (Liu et al., 2023b) employs a voxel-based sparse 3D-CNN to process point clouds and regress on quality scores. MM-PCQA (Zhang et al., 2022c) utilizes a multi-modal learning approach on point cloud patches and image projections, where cross-modal attention is used to fuse image and point cloud features, upon which a quality score is regressed. Nevertheless, all these approaches are highly dependent on the availability of labeled data, and thus have poor generalization once tuned.

Several approaches have been tried to reduce the dependence on labeled data. IT-PCQA (Yang et al., 2022a) uses unsupervised domain adaptation to transfer the power of deep quality evaluators designed for the 2D case to the 3D setting, but the significant domain gap leads to unsatisfactory performance. Recently, the release of LS-PCQA (Liu et al., 2023b), a relatively large-scale dataset for point cloud quality assessment, has opened up avenues for pretraining, similar to the 2D case. As discussed earlier, contrastive learning has been employed with great success in image and video quality assessment. CONTRIQUE (Madhusudana et al., 2022) trains similarity under quality-preserving augmentations to obtain robust features useful for quality assessment. In the 3D space, CoPA (Shan et al., 2024) takes image projections of point clouds and mixes patches from them to create positive pairs. As mentioned earlier, while contrastive approaches produce robust and useful features, they require a complex and expensive positive and negative pair generation process, which limits their application on even larger datasets.

Outside the quality space, multimodal large language models have shown impressive performance in multimodal understanding, be it text-image (Liu et al., 2023a), text-audio (Huang et al., 2023), text-motion (Jiang et al., 2023) etc. A key idea behind progress has been that of visual instruction fine-tuning as proposed in LLaVA (Liu et al., 2023a), which shows that LLMs can be made to understand visual content with a relatively simple two-stage fine-tuning scheme. Point-LLM (Xu et al., 2023) proposes LLaVA-style instruction fine-tuning for point clouds, where it leverages a pretrained point cloud encoder and projector to inject point cloud features and perform multimodal instruction fine-tuning. PointBind (Guo et al., 2023) uses a contrastive learning approach to align point cloud features with other modality features, allowing their downstream use in other applications. However, all of these methods are designed with high-level semantic tasks like object classification in mind, so cannot handle the large-scale point clouds in PCQA well.

In the 2D space, Q-Bench (Wu et al., 2023a) demonstrated that LMMs not only understand high-level semantics, but also low-level details relevant for quality assessment. Wu et al. (2024) perform an evaluation of LMMs for quality assessment, conducting ablations to figure out which setting performs the best. Building on this, Q-Align finetunes an off-the-shelf vision-language LMM on discretized quality scores and converts predicted logits to MOS using softmax pooling. However, we show that naively applying Q-Align to the 3D case results in a significant domain gap, likely due to the information loss involved in moving from the 3D to the 2D case. Our proposed method therefore introduces the point cloud modality to the mix, and demonstrates that the advantages obtained by LMM-based quality models also hold for the 3D case when the relevant information is available.

LMM-PCQA (Zhang et al., 2024) builds on Q-Align by first fine-tuning it on cubic image projections, and then training a regressor on predicted logits from the fine-tuned image-text model along with classical point cloud structural features such as linearity and planarity. As discussed earlier, this approach is limited in the way it extracts information from the point cloud, and cannot leverage the unique power of multimodal training to learn interactions between the different data modalities. On the other hand, our approach seamlessly integrates deep point cloud encoders with existing image-text foundation models, thus enabling true end-to-end multimodal training.

## 3 METHOD

In this section, we first describe the construction of our instruction-following dataset, where we compose inputs of multiple modalities to produce an input prompt. We then delve into the architecture of PIT-QMM, which takes in text, images and point clouds to produce a quality score. Finally, we detail the label discretization and smoothing strategy and our two-stage training recipe.

### 3.1 POINT-PROJECTION-TEXT INSTRUCTION-FOLLOWING QUALITY DATA

#### 3.1.1 POINT CLOUDS

As discussed, the most challenging aspect of including point clouds in the input for this task is the relatively large point cloud size in standard quality assessment datasets. Popular point cloud encoders such as Point-BERT (Yu et al., 2022), Point-MAE (Pang et al., 2022) and I2P-MAE (Zhang et al., 2023) are usually pretrained on ShapeNet (Chang et al., 2015) or Objaverse (Deitke et al., 2023), which contains point clouds containing thousands of points, whereas quality assessment datasets such as LS-PCQA and WPC contain point clouds with hundreds of thousands to millions of

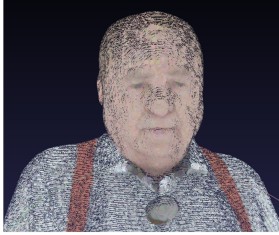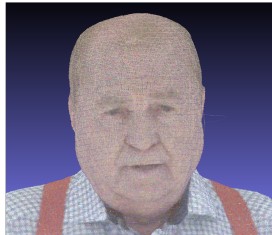

Figure 2: The same underlying point cloud can have highly different quality characteristics depending on rendering parameters and the radius of interaction, especially in the NR setting. Point cloud taken from LS-PCQA.

points. This is to be expected, as high-frequency quality impairments can only become apparent in point clouds with sufficient granularity. However, this means that we cannot simply feed the entire point cloud into the input, as pretrained point cloud encoders are not optimized to process inputs of this size, even if they are even able to consume them. Moreover, we are also limited by the finite context length of the LMM, so cannot naively break the larger point cloud into smaller chunks and pass them all in sequentially.

As a result of this, we sample the point cloud in three ways to provide three different levels of information. First, we apply furthest point sampling to the cloud to obtain an overall sparse view. This view represents the broad shape of the cloud to provide content-level information, which can guide the final quality score, as in ReIQA. Next, we randomly sample a small set of local patches from the point cloud to provide information about local variations, such as high-frequency distortions. These two views together form our point cloud inputs to the model. We also explore a variant where we take patches at two different scales, analogous to the two-scale inputs typically observed in 2D quality assessment. Here, we have patches from the original point cloud and a uniformly downsampled version of the point cloud, where we downsample by a factor of 2. We report the performance of this variant in the ablation study. Note that these views comprise a small fraction of the total number of points in the cloud (typically 3-5%). While processing such a small number of points makes our method much more efficient as compared to processing the entire cloud, it also means that we need complementary global information to capture a more holistic picture of the point cloud.

### 3.1.2  IMAGE PROJECTIONS

As discussed in Section 3.1.1, since we can only provide a limited set of local views with point clouds due to model constraints, we also add image projections of the point cloud to our input. For a point cloud $P$, we normalize it to zero-mean and unit-maximum distance with $\mathcal{N}(\cdot)$, then render $\mathcal{N}(P)$ into multi-view images $\{x_i \in \mathbb{R}^{H \times W \times C}|_{i=1}^{6}\}$ from six perpendicular viewpoints (i.e., along the positive and negative directions of $x, y, z$-axes) with fixed viewing distance. These projections not only provide a global view of the point cloud, but also allow us to leverage the power of pretrained large image quality models. We set the projection parameters as used by the original dataset authors wherever available, else apply best effort settings as appropriate.

### 3.1.3  TEXT

In the text portion of the prompt, we first prime the LMM for quality assessment. Specifically, we state the task as no-reference quality assessment with single-stimulus absolute category ratings. Besides priming, this also provides information about the psychometric aspect of the dataset which we show to be useful guidance for the LMM. Moreover, we observe that quality assessment for point clouds is highly dependent on the settings used to render the point cloud and how the user was allowed to interact with it. For example, Figure 2 shows the same point cloud rendered with different point sizes and viewing distances, all of which have significantly different quality characteristics. This is a complexity typically not observed in 2D quality datasets. Accordingly, we also include rendering parameters in our prompt as described in the corresponding datasets when available or a best effort reproduction when not.

Table 1: **Instruction following prompt.** {System Prompt} is the system prompt used by the pre-trained LLM, {Experimental Setup} is a description of the psychometric experimental setup, {im_tokens} are image tokens and {p_tokens} are point tokens.

| {System Prompt} | |
| --- | --- |
| USER: | This is a point cloud rated for quality. It was displayed to a human in a single stimulus setup with absolute category ratings. {Experimental Setup}{im_tokens}<p_start>{p_tokens}<p_end> Can you rate the quality of the point cloud? |
| ASSISTANT: | The quality of the point cloud is excellent. |

### 3.1.4 Final Instruction-Following Prompt

An example final question-answer input pair is in Table 1. Similar to Point-LLM, we also add the special tokens `<p_start>` and `<p_end>` to demarcate the start and end of the point cloud. Note that we use discrete quality levels in our output, as discussed in more detail in Section 3.3.1.

## 3.2 Model Architecture

As shown in Figure 1, our PIT-QMM is a generative model that aims to complete multi-modal sentences containing point clouds, images and text. The model consists of four main components - an image encoder $f_{im}$, a point cloud encoder $f_{point}$, a point cloud embedding projector $f_{point\_proj}$, and a large language model (LLM) backbone $f_{llm}$.

The point cloud encoder $f_{point}$ takes in a point cloud $P \in \mathbb{R}^{s \times n \times d}$, where $s$ is the number of sampled views, $n$ is the number of points and $d$ is the feature dimension. The output is a sequence of patched point features $X \in \mathbb{R}^{s \times m \times c}$, where $m$ is the number of patch features and $c$ is the feature dimension. The projector $f_{proj}$ is a multi-layer perceptron (MLP) that maps the point features $X$ to point tokens $Y \in \mathbb{R}^{s \times m \times c'}$, where $c'$ is the dimension of the point tokens, which is the same as the text and image tokens. Finally we flatten this into a sequence $Z \in \mathbb{R}^{sm \times c'}$, which we feed into $f_{llm}$.

The LLM $f_{llm}$ takes in a sequence of the form $\mathbb{R}^{n' \times c'}$, where $n'$ is the length of the mixed image, text and point cloud input token sequence. As a decoder-only LLM, it produces a probability distribution of predictions for the next token of size $\mathbb{R}^V$ for a given input sequence, where $V$ is the vocabulary size of the LLM, and autoregressively produces the output sequence.

The image encoder $f_{im}$ and LLM backbone $f_{llm}$ in our implementation are architecturally similar to the open source LMM mPLUG-Owl-2 (Ye et al., 2024). In addition to its base image encoder, mPLUG-Owl-2 also has a visual abstractor that greatly reduces the number of tokens needed to represent an image, thus allowing us to process all six cubic views efficiently. Moreover, it features modality-aware modules that allow learning the interactions between different modalities effectively. However, note that the overall architecture is agnostic to the kind of image encoder, point cloud encoder and LLM used, and these may be easily replaced with any off-the-shelf variant as desired.

## 3.3 Training and Inference

The model is trained end-to-end by minimizing the negative log-likelihood of the token at each position. We ignore the image tokens, point cloud tokens, user prompt, and instruction tokens for computing the loss, so that the model can focus on producing relevant and coherent responses.

### 3.3.1 Label Smoothing and Discretization

As observed in Q-Align, LMMs optimized for quality perform better when they are asked to produce discrete text labels, largely due to their bias to produce text as opposed to numeric values. As a result, we follow a similar discretization strategy during training, where we convert continuous quality scores to five-point Likert scale levels in our input prompts. The labels are equally spaced based on the score ranges in the respective quality datasets. During inference, we convert the discrete labels into continuous quality scores by first assigning them discrete numeric levels (e.g. 1 to 5) and then

taking a weighted average of the numeric levels based on the corresponding token probabilities in the output of the LLM.

### 3.3.2 Two-stage Training

As in LLaVA (Liu et al., 2023a), we also employ a two stage training strategy. In the first feature alignment stage, we freeze all parameters except for the point cloud projector, and train on brief-description instructions from the Cap3D (Luo et al., 2023) dataset, which is for 3D captioning. As all the point clouds in this dataset are relatively small, we do not apply our point cloud sampling strategy at this step. This stage involves aligning the point cloud features with the image and text features.

In the second instruction-tuning stage, we unfreeze the image abstractor and add LoRA adapters to the LLM and the point cloud encoder. We then finetune the weights end-to-end using the constructed quality dataset. In this stage, all the modules are tuned specifically for point cloud quality assessment. For the image abstractor, this involves adapting to the distribution shift of synthetic point cloud projections as opposed to the general space of images, whereas the point cloud encoder has to adapt to the distribution of local patches with high-frequency variations instead of object-level point clouds.

## 4 Experiments

### 4.1 Datasets

Our experiments are based on three popular point cloud quality assessment datasets, namely LS-PCQA (Liu et al., 2023b), SJTU-PCQA (Yang et al., 2021), and WPC (Liu et al., 2022). LS-PCQA is a large-scale point cloud quality assessment dataset with 104 pristine and 24,024 distorted point clouds. Each pristine point cloud is impaired with 33 types of distortions under 7 levels. The labels in LS-PCQA are mostly synthetically generated pseudo-MOSs, with only 930 samples having psychometrically-collected true MOSs. We term this subset LSPCQA-small and report results of ablations on it, along with WPC. SJTU-PCQA contains 9 reference and 378 distorted samples impaired with 7 types of distortions under 6 levels, while WPC contains 20 reference point clouds and 740 distorted sampled disturbed by 5 types of distortions.

### 4.2 Evaluation Protocol

We tested our PIT-QMM model against other state-of-the-art models on all of the datasets described in Section 4.1. We first constructed instruction-tuning data from the raw datasets as described in Section 3.1. Each sample is then a set of point cloud samples, cubic image projections and user-agent instruction text. We split each dataset into content-separated train-test sets in the ratio of 4:1. We then minimized loss on the training set and obtained metrics on the test set. Due to the stochasticity involved in sampling from the point cloud, we computed metrics on the test set with 10 different seeds and took the mean. Finally, the test metrics were averaged over 5 different train-test splits to obtain the final reported metrics. Two widely adopted evaluation metrics were employed to quantify the level of agreement between predicted quality scores and MOSs: Spearman rank order correlation coefficient (SROCC) and Pearson linear correlation coefficient (PLCC).

### 4.3 Implementation Details

The point cloud projections were rendered with PyTorch3D (Ravi et al., 2020) at a resolution of $512 \times 512$. All point cloud samples are $n = 8192$ dimensional with 3 spatial coordinates and 3 RGB color coordinates, which makes $d = 6$. The furthest point sampling was done with the Python package fpsample with the bucket-based FPS algorithm (Han et al., 2023). To sample local patches, we constructed a search tree using the Python package FAISS, sampled a single point randomly and then looked up the closest points near it to construct the final sample. We sampled three patches in total including the furthest point sample of the point cloud.

Our experiments were performed with Huggingface and PyTorch (Paszke et al., 2019) using $3 \times 40$ GB NVIDIA A100 GPUs. For the point cloud encoder, we used Point-BERT pretrained with ULIP-

2 (Xue et al., 2024) on the Objaverse (Deitke et al., 2023) dataset. The point encoder outputs $m = 513$ point features, each with $c = 384$ dimensions. The point cloud projector is taken as a randomly initialized MLP. The projector contains three linear layers with the GeLU (Hendrycks & Gimpel, 2016) activation, which maps point features to tokens with $c' = 5120$ dimensions. The image encoder is taken as a Vit-L/14 (Ilharco et al., 2021) and the LLM is taken from mPLUG-Owl2, which is a modified LLaMA-2-7B (Touvron et al., 2023) model. Since we added two additional special tokens, the vocabulary size of PIT-QMM is $V = 32003$. The weights of the image encoder and LLM were initialized from Q-Align.

For the alignment stage, we pretrained on the instruction-following variant of Cap3D as released in Point-LLM (Xu et al., 2023) for 3 epochs with a batch size of 12. We used a learning rate of $2 \times 10^{-3}$ with cosine annealing and a warmup rate of 0.3. All other hyperparameters were the same as those used in Point-LLM. In the finetuning stage, we trained on LS-PCQA for 5 epochs, SJTU-PCQA for 90 epochs and WPC for 30 epochs. We used a learning rate of $2 \times 10^{-4}$ with cosine annealing and a warmup rate of 0.3. For the LoRA (Hu et al., 2021) modules, we used $r = 128$, $\alpha = 256$, and dropout of 0.05 on the multiway $V_{proj}$ and $Q_{proj}$ layers in mPLUG-Owl2, and the $V$ and $Q$ matrices in Point-BERT.

## 4.4 COMPARISON WITH STATE-OF-THE-ART METHODS

We selected 16 state-of-the-art PCQA methods for comparison, including 10 FR-PCQA and 6 NR-PCQA methods. The FR-PCQA methods include MSE-p2point (Mekuria et al., 2016), Hausdorff-p2point (Mekuria et al., 2016), MSE-p2plane (Tian et al., 2017), Hausdorff-p2plane (Tian et al., 2017), PSNR-yuv (Torlig et al., 2018), PointSSIM (Alexiou & Ebrahimi, 2020), PCQM (Meynet et al., 2020), GraphSIM (Yang et al., 2020), MS-GraphSIM (Zhang et al., 2021), and MPED (Yang et al., 2022b). The NR-PCQA methods include PQA-Net (Liu et al., 2021), IT-PCQA (Yang et al., 2022a), GPA-Net (Shan et al., 2023), ResSCNN (Liu et al., 2023b), MM-PCQA (Zhang et al., 2022c), and CoPA+FT (Shan et al., 2024). We did not compare against 3DTA (Zhu et al., 2024) since their evaluation protocol differs significantly from ours, as they use only a single test-train split for computing their metrics. We also did not include LMM-PCQA as they do not provide all of their code and the test-train splits used for evaluation, and hence their scores are not reproducible. Moreover, they do not report scores on LS-PCQA, the largest and most challenging of popular PCQA databases.

### 4.4.1 WITHIN-DATASET PERFORMANCE

The within dataset performance on LS-PCQA, SJTU-PCQA and WPC is reported in Table 2. From the table, we have the following observations: 1) Our model outperformed all the NR-PCQA methods on all three datasets. For example, it outperforms the current state-of-the-art by a margin of 22.5% in SROCC and 20.4% in terms of PLCC. 2) Our model also outperformed all the FR PCQA methods on all the three datasets. Here the improvement is not as dramatic due to FR methods having the access to the pristine source, but the gap is nevertheless significant, especially due to the large disparity in available information. 3) Our model delivers robust performance across the three datasets, despite variations dataset scale, content, and distortion types.

### 4.4.2 TRAINING COST

As demonstrated in Table 3, PIT-QMM converges to best results when tuning for quality with fewer epochs as compared to other state-of-the-art learning-based methods. We report these verbatim from the respective technical reports or the code provided if not available in the reports. We observe that the savings are most significant on the large LS-PCQA dataset, where merely 5 epochs are sufficient to obtain state-of-the-art performance. On the other hand, on the much smaller SJTU-PCQA dataset, we require more iterations, likely due to the larger number of parameters to be tuned. This downstream training efficiency is a relatively common phenomenon when using foundation models, which demonstrate impressive zero-shot and few-shot capabilities (Brown, 2020).

Table 2: Performance results on the LS-PCQA (Liu et al., 2023b), SJTU-PCQA (Yang et al., 2021) and WPC (Liu et al., 2022) databases. "P" and "I" stand for the method is based on the point cloud and image modality, respectively. ↑ indicates that larger is better. The best performance results are marked in **RED** and the second results are marked in **BLUE** for both FR-PCQA and NR-PCQA methods. "FT" indicates fine-tuning.

| Ref | Modal | Methods | LS-PCQA | | SJTU-PCQA | | WPC | |
|-----|-------|---------|---------|---------|-----------|---------|-----|-----|
| | | | SROCC ↑ | PLCC ↑ | SROCC↑ | PLCC↑ | SROCC↑ | PLCC↑ |
| FR | P | MSE-p2po | 0.325 | 0.528 | 0.783 | 0.845 | 0.564 | 0.557 |
| | P | HD-p2po | 0.291 | 0.488 | 0.681 | 0.748 | 0.106 | 0.166 |
| | P | MSE-p2pl | 0.311 | 0.498 | 0.703 | 0.779 | 0.445 | 0.491 |
| | P | HD-p2pl | 0.291 | 0.478 | 0.617 | 0.661 | 0.344 | 0.380 |
| | P | PSNR-yuv | **0.548** | **0.547** | 0.704 | 0.715 | 0.563 | 0.579 |
| | P | PointSSIM | 0.180 | 0.178 | 0.735 | 0.747 | 0.453 | 0.481 |
| | P | PCQM | 0.439 | 0.510 | 0.864 | 0.883 | **0.750** | **0.754** |
| | P | GraphSIM | 0.320 | 0.281 | 0.856 | 0.874 | 0.679 | 0.693 |
| | P | MS-GraphSIM | 0.389 | 0.348 | **0.888** | **0.914** | **0.704** | **0.718** |
| | P | MPED | **0.659** | **0.671** | **0.898** | **0.915** | 0.656 | 0.670 |
| NR | I | PQA-Net | 0.588 | 0.592 | 0.659 | 0.687 | 0.547 | 0.579 |
| | I | IT-PCQA | 0.326 | 0.347 | 0.539 | 0.629 | 0.422 | 0.468 |
| | P | GPA-Net | 0.592 | 0.619 | 0.878 | 0.886 | 0.758 | 0.76 |
| | P | ResSCNN | 0.594 | 0.624 | 0.834 | 0.863 | 0.735 | 0.752 |
| | P+I | MM-PCQA | 0.581 | 0.597 | 0.876 | 0.898 | 0.761 | 0.774 |
| | P | CoPA+FT | **0.613** | **0.636** | **0.897** | **0.913** | **0.779** | **0.785** |
| | P+I | **PIT-QMM** | **0.751** | **0.766** | **0.911** | **0.923** | **0.824** | **0.793** |

Table 3: Epochs required to converge to best results across all databases. Bold denotes the best performing model.

| Method | Batch size | LS-PCQA | SJTU-PCQA | WPC |
|--------|-----------|---------|-----------|-----|
| PQA-Net | 20 | 160 | 160 | 160 |
| MM-PCQA | 8 | 50 | **50** | 50 |
| CoPA + FT | 16 | 20 | 150 | 150 |
| PIT-QMM | 10 | **5** | 90 | **30** |

## 4.5 ABLATION STUDY

To study the effectiveness of our proposed dataset construction strategy, we conducted an ablation study to investigate the individual contribution of different components of each training sample. Table 4 summarizes the results of this study. We used the WPC and LSPCQA-small databases in these ablations.

First, we report the performance of using only 2D image projections to predict quality (row ①). We constructed our datasets exactly as before, except for the point clouds, and tune Q-Align on them, following the recipe recommended by the authors. For fairness, we trained for the same number of iterations as we did in PIT-QMM. We observed that a tuned Q-Align model was able to achieve state-of-the-art performance on LSPCQA-small and WPC, though the margin of outperformance was lower, especially when compared to the FR algorithms. This validates the inclusion of point cloud projections and highlights the value of using pretrained vision foundation models in this domain.

Next, we investigated the effectiveness of different types of point cloud sampling. We considered three schemes. In the first, we sampled only local patches from the point cloud at full scale (row ②). In the second, we added a furthest point sample along with local patches (row ③). In the third, we added patches at half-scale resolution along with the aforementioned (row ④). The motivation behind this final experiment is the observation that multi-scale processing in 2D computer vision is often beneficial, even for quality assessment (Wang et al., 2003). For the half-scale point cloud, we uniformly downsampled the point cloud by a factor of 2.

Our first observation was that any kind of point cloud sampling improved performance over the baseline of using 2D projections only, which validates the inclusion of point clouds into the train-

Table 4: Ablation study on the LSPCQA-small (Liu et al., 2023b) and WPC (Liu et al., 2022) databases. ↑ indicates that larger is better.

| Methods | LSPCQA-small | | WPC | |
|---|---|---|---|---|
| | SROCC↑ | PLCC↑ | SROCC↑ | PLCC↑ |
| ① | 0.684 | 0.664 | 0.781 | 0.687 |
| ② | 0.722 | 0.681 | 0.793 | 0.755 |
| ③ | 0.734 | 0.699 | 0.819 | 0.774 |
| ④ | 0.730 | 0.694 | 0.812 | 0.769 |
| ⑤ | 0.343 | 0.322 | 0.447 | 0.405 |
| ⑥ | 0.733 | 0.704 | 0.824 | 0.793 |
| ⑦ | 0.737 | 0.706 | 0.822 | 0.790 |

ing. Next, we observe that only sampling local patches yielded the worst overall results. This is somewhat expected, as there is a significant domain gap for our pretrained point cloud encoder to overcome. The encoder was trained to obtain an overall semantic understanding of object-like point clouds, not to understand the fine-level details of patches of point clouds. Adding a furthest point sample improved on the local patch-only case. As discussed earlier, the furthest point sample would be processed into a content-oriented feature by the point cloud encoder, given that this aligns well with its pretraining task. This improvement therefore ties in with observations made elsewhere that content-oriented features provide complementary information for quality assessment. Lastly, we notice that adding another scale of information did not change the results by much. We note that this strategy may be more effective if the low resolution and high resolution patches were paired, thus allowing the model to learn bandpass features, and leave this investigation for future work.

Another test we performed is to use only the point cloud features to predict quality (row ⑤). In this case, we simply dropped the image tokens and trained end-to-end as usual. The performance here dropped significantly, thus providing evidence that the point cloud encoder by itself has to overcome a large domain gap to provide sufficiently discriminative features for quality assessment. Finally, we investigated the importance of text conditioning. We trained with three varieties of prompts. In the first, we trained with minimal context, where we simply asked the question i.e, *Q: Can you predict the quality of the point cloud?*. This is the same as row ④. In the second, we provided task priming as well as psychometric information i.e, *Q: This is a point cloud rated for quality. It was displayed to a human in a single-stimulus setup with absolute category ratings. Can you predict the quality of the point cloud?* (row ⑥). In the last, we also included rendering parameters in the text prompt, such as the distance of the cameras and projection type (row ⑦).

Here, we found that each variety improves slightly on the baseline, but the performance was overall comparable. Hence, it appears explicitly stating the task and experimental conditions does guide the LMM towards producing better quality predictions, but the difference might easily be overcome through other means. We posit that text conditioning will be more useful when the LMM is being trained for a multi-task or multi-modality quality task, such as FR and NR together, or FR on video and point clouds jointly. This conditioning would allow it to specialize to the specific sample at hand. However, since this exploration is beyond the scope of this paper, we leave this investigation for future work.

## 5 CONCLUSION

In this paper, we presented a novel end-to-end no-reference point cloud quality assessment algorithm based on LMMs. By leveraging complementary sources of information from different modalities and the power of large pretrained modality encoders, the proposed PIT-QMM is able to predict quality scores across a wide variety of distortion and content types. Extensive experiments show that PIT-QMM is able to achieve competitive performance across a wide variety of benchmarks with overall fewer training iterations than state-of-the-art methods. Thorough ablations also validated each step of our dataset construction strategy.

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
