# OpenReview forum: "PIT-QMM: A Large Multimodal Model for No-Reference Point Cloud Quality Assessment"
_ICLR.cc/2025/Conference — ICLR 2025 Conference Withdrawn Submission_

### Official Review · Reviewer_HzwC · 2024-10-16

**Soundness:** 3
**Presentation:** 2
**Contribution:** 2
**Rating:** 5
**Confidence:** 5

**Summary:**

This paper proposes PIQ-QMM, which leverages multimodal information by jointly extracting features from the perspectives of text, 2D projections, and 3D point clouds to regress into a quality score. Overall, the idea is rather conventional and can be seen as incremental work. There are also some issues with the writing and experiments.

**Strengths:**

1. The introduction of related works on `Point Cloud Quality Assessment` is relatively clear, providing a detailed explanation of the differences between the proposed method and previous approaches.
2. The description of the proposed method is clear, allowing readers to quickly grasp the core idea behind it.

**Weaknesses:**

1. The motivation for the method is limited. PIQ-QMM can be simply viewed as using an image encoder, point cloud encoder, and tokenizer to extract features from three modalities, then letting the LMM analyze and regress the quality score. There is no strong theoretical explanation of how these modalities influence each other, which weakens the motivation. Adding necessary visualizations in the introduction section and analyzing potential influences between the modalities would make the argument more convincing.
2. The framework diagram in the paper is too simplistic and does not fully and clearly present the proposed method. For example, Figure 1 is drawn too simply, causing the loss of important information.
3. There are obvious issues in the experiment section. In Section 4.1, the authors mention that SJTU-PCQA has nine models, and in Section 4.2, they claim that the train-test split is 4:1, which is a factual error. Additionally, in the datasets used by the authors, such as WPC and SJTU-PCQA, there are only 378 and 740 trainable point clouds, respectively. Is it reasonable to train large models with such a small amount of data, and could this lead to overfitting? The authors should clarify this point.

**Questions:**

In summary, this paper is a good attempt to build upon previous work and verifies the feasibility of the idea. However, the motivation is not clearly expressed, the writing needs improvement, and the experiment section contains factual errors and lacks necessary explanations.
The authors can refer to the weaknesses of the questions.

---

### Official Review · Reviewer_i1hN · 2024-10-29

**Soundness:** 3
**Presentation:** 2
**Contribution:** 2
**Rating:** 5
**Confidence:** 4

**Summary:**

The study addresses the challenge of assessing the perceptual quality of point clouds, which are 3D representations widely used in applications like autonomous driving and digital twins.  The authors propose PIT-QMM (Point-Image-Text Quality Multimodal Model), which integrates multiple modalities (text, images, and 3D point clouds) into a unified architecture. PIT-QMM achieves state-of-the-art performance on popular benchmarks such as LS-PCQA, SJTU-PCQA, and WPC, outperforming other methods with fewer training iterations.

**Strengths:**

- The paper proposes the PIT-QMM model, which bridges the gap between 2D and 3D quality assessment through a point-image-text multimodal approach.
- The experiments are extensive, with results demonstrating that the model achieves state-of-the-art performance across various point clound quality assessment benchmarks.

**Weaknesses:**

The novelty of this paper is questionable. First, the use of LMM to evaluate the quality of images and videos has already been validated in prior studies, such as Q-Align. Therefore, applying a similar approach to assess the quality of point clouds does not offer incremental contributions to the field. Moreover, the training method used to derive the quality scores for point clouds—an essential element of applying LMM to quality assessment—remains identical to that of Q-Align. Second, LMM-PCQA adopts a similar approach (with only slight differences in point cloud feature extraction), which was already published in ACM MM 2024. Third, the multi-modal feature fusion of 2D projections and 3D point clouds also closely resembles the approach used in MM-PCQA.

**Questions:**

- The authors mention that LS-PCQA primarily consists of synthetically generated pseudo-MOSs. Is it appropriate to use it as a test set to validate the performance of quality assessment methods in Table 2?
- The performance of MM-PCQA on the SJTU-PCQA and WPC datasets is weaker than reported in its original paper. The authors should state the reason.
- The paper does not compare the proposed method with LMM-PCQA, the most similar approach. Although the authors state that the full code and specific test-train splits used for evaluation are not provided, I believe that if the training splits are comparable, the results are still meaningful and should be referenced.

---

### Official Review · Reviewer_cdaM · 2024-11-03

**Soundness:** 2
**Presentation:** 2
**Contribution:** 1
**Rating:** 3
**Confidence:** 5

**Summary:**

This paper presents an LMM-based NR-PCQA method that is capable of consuming text, images and point clouds to predict quality scores.

**Strengths:**

An LMM-based NR-PCQA method is proposed, on the basis of LMM-based NR IQA/VQA methods.
A dataset construction recipe is introduced to help the model training.
The proposed method achieves good performances.

**Weaknesses:**

Both novelty and contribution of the paper are limited.
The introduced model, database construction, and model design are all based on the existing LMM-based IQA and VQA studies. Only small adaptions are made to adapt the format of 3D point clouds, which heavily limits the novelty and contribution of the paper.
The experimental validation is also very limited. The scales of the databases are all very small. Only several small tables of quantitative results are given.

**Questions:**

See comments above.

---

### Official Review · Reviewer_qaQW · 2024-11-03

**Soundness:** 3
**Presentation:** 3
**Contribution:** 2
**Rating:** 3
**Confidence:** 5

**Summary:**

The paper presents PIT-QMM, a novel Large Multimodal Model designed for No-Reference Point Cloud Quality Assessment (NR-PCQA). Recognizing the limitations of existing methods in evaluating the perceptual quality of point clouds without a reference, the authors propose a comprehensive approach that integrates multiple modalities—specifically, text descriptions, 2D projections, and 3D point cloud views.

Key contributions include:
(1)PIT-QMM Development: A pioneering multimodal model that combines text, image, and point cloud data to predict quality scores.
(2)Innovative Dataset Strategy: A robust methodology for constructing a dataset that enhances learning and generalization for point cloud quality assessment.
(3)Empirical Validation: Extensive experiments show that PIT-QMM outperforms state-of-the-art methods with fewer training iterations, supported by comprehensive ablation studies confirming the effectiveness of its components.

**Strengths:**

1.The methodology is robust, with a clear design and execution.
2.The authors provide a comprehensive dataset construction strategy that enhances the model's learning capacity.

**Weaknesses:**

1.The motivation behind such a LLM-based design is not clearly explained. In fact, utilizing information of different modalities for PCQA has been widely explored in previous works. Just replacing traditional image/point cloud encoders with a LLM lacks enough novelty.
2. The experimental validation lacks cross-dataset verification.

**Questions:**

Please refer to the weaknesses.

---

### Official Review · Reviewer_gKko · 2024-11-03

**Soundness:** 3
**Presentation:** 3
**Contribution:** 2
**Rating:** 5
**Confidence:** 3

**Summary:**

The work proposed the first end-to-end point-image-text multimodal model optimized for PCQA, namely PIT-QMM. This approach is capable of consuming text, images, and point clouds to predict quality scores, which outperform several PCQA method with fewer training iterations.

**Strengths:**

This article provides an anonymous repository with complete code and strong reproducibility.

This article uses the two-stage training method to implement instruction tuning and notes the training details.

**Weaknesses:**

This article spends a lot of space describing LLM-PCQA, but does not compare it in experiments, which is very imprecise. As a QA method based on LLM, the author cannot just state that LLM-PCQA uses feature fusion and PIT-QMM uses instruction tuning. It is necessary to compare the performance in detail.

The model is trained in a very similar way (two-stage) to Q-Align's predecessor, Q-Instruct, but the author does not mention it.

**Questions:**

I checked the author's anonymous GitHub, which contains content from many datasets, especially two-dimensional ones, but many of them are not cited in the text. I don't know if it is a mistake due to improper citation, or if the data is not used. Considering the point-image-text multimodal model claimed by the author, I assume that this data is used during training. If not used, why does it appear in the dataset (such as test_jsons)?

---

### Note · Authors · 2024-11-22

I have read and agree with the venue's withdrawal policy on behalf of myself and my co-authors.